# Fine-Scale Distribution Patterns of *Phragmites australis* Populations Across an Environmental Gradient in the Salt Marsh Wetland of Dunhuang, China

**Liang Jiao \*, Fang Li, Xuerui Liu, Shengjie Wang and Yi Zhou**

College of Geography and Environmental Science, Northwest Normal University, Lanzhou 730070, China; lifang126385@163.com (F.L.); liuxuerui187@163.com (X.L.); geowangshengjie@163.com (S.W.); zhouyihhhi@163.com (Y.Z.)

\* Correspondence: jiaoliang@nwnu.edu.cn; Tel.: +00-86-1391-9350-195

**Abstract:** The spatial distribution pattern of plants often reflects their ecological adaptation strategy and is formed by their long-term interaction with environmental factors. In this paper, the clonal plant, *Phragmites australis*, was investigated across environmental gradients, including the wet zone, the transitional zone and the desert zone of the salt marsh wetland of Dunhuang, China. The characteristics and influencing factors of their fine-scale spatial distribution patterns were studied by point pattern analysis, redundancy analysis and simple linear regression. The results show that: (1) the spatial distribution pattern of *Phragmites australis* changes from aggregation to non-aggregation (random and regular distribution) from the wet zone to the desert zone. (2) The soil water content, pH and salinity all affect *Phragmites australis'* spatial distribution intensity. Simple linear regression reveals that the water content in each soil layer, the pH of the deep soil layer and the salinity of the surface and deep soil layers are the main soil conditions of *Phragmites australis'* spatial distribution pattern. (3) *Phragmites australis'* population characteristics and clonal characteristics also have significant effects on its spatial distribution intensity. Specifically, the intensity of its spatial distribution pattern is significantly positively correlated with its cover, frequency, density, height, biomass, node number, ramet number and stem diameter ($p < 0.01$), while it is significantly negatively correlated with its rhizome internode length, spacer length and branch angle ($p < 0.01$). This research clarified the relationship between the spatial distribution pattern of *Phragmites australis* with soil environmental factors, plant clonal characteristics and population characteristics. The results provide a theoretical basis for understanding the ecological adaptation mechanism of clonal plants and protecting the sustainability of fragile and sensitive inland river wetland ecosystems.

**Keywords:** salt marsh wetland; clonal plant; *Phragmites australis*; spatial point pattern; ecosystem sustainability

## 1. Introduction

Clonal plants produce genetically identical progeny by vegetative propagation under natural conditions. They have strong environmental adaptability due to a series of life history traits, such as clonal plasticity, clonal integration, division of labor, foraging behavior and coordinate resource allocation among organs and tissues [1–3]. Clonal plants are widely distributed in various ecosystems and play an important ecological role, especially in wetlands, grasslands, waters and other special and fragile ecosystems [4,5]. There exists a discrepancy in the spatial adaptability of clonal plants and non-clonal plants due to their different intrinsic nature, spatial mobility, method of reproduction and resource sharing [6–8]. However, related studies mostly focus on non-clonal plants and pay little

attention to clonal plants [9,10]. Therefore, the study of the spatial distribution patterns of clonal plants can complement and perfect the theory and practice of clonal plant ecology.

Arid and semi-arid regions account for about 40% of the global land surface area and show an increased risk of land degradation and desertification under global warming and drought conditions [11]. Compared with coastal wetlands and lake and river wetlands, salt marsh wetlands in arid and semi-arid regions have a lower environmental capacity, lower ecological carrying capacity and a more fragile ecosystem. However, salt marsh wetland, as the basis of desert oasis development, plays an essential role in maintaining global ecological balance and protecting the earth ecosystem [12]. *Phragmites australis*, as the dominant and constructive species of salt marsh wetlands, can help to maintain the stability of wetland ecosystems. With clonal reproduction, it shows a strong phenotypic plasticity and ecological adaptability [13–15]. However, current studies on *Phragmites australis* mainly focus on their physiological and biochemical characteristics [16–18], biological responses to environmental factors [19,20], the trade-offs between vegetative and sexual reproduction [17,21], intraspecies relationships [22], ecosystem function and economic value [23–25]. There are few studies on the spatial distribution pattern of *Phragmites australis* in salt marsh wetlands, which could prevent us from understanding the ecological adaptation strategies of clonal plants.

Resource heterogeneity controls species diversity and population spatial distribution pattern [26]. Population spatial distribution pattern plays an important role in constituting population morphology and structure, community succession and ecosystem evolution, and usually shows a certain predictability [27,28]. At present, the methods of studying population spatial patterns include hierarchical analysis of variance, species-juxtaposition and point pattern analysis. Among them, point pattern analysis has been widely used for studying the spatial distribution patterns of plant populations [9,29], animal populations [30,31] and geographical landscapes [32,33].

The heterogeneity of resources and other environmental conditions is often manifested as patchiness in space [34]. Spatial distribution patterns of plant populations are closely linked to the scale of patches used for investigation. For example, the spatial distribution patterns of three dominant shrub species in semi-arid Patagonia are aggregated on a 1.1 m scale but are random on other scales [35]. Researchers found that the changes of plant weight and fecundity can be better explained by neighborhood individuals within distances of 2 cm and 5 cm, respectively, when modeling the competition among annual herbaceous plants [36,37]. Comparison of the spatial distribution patterns of different plant species reveal that the interaction between herbaceous plants often occurs in the range of centimeters. Fine-scale spatial distribution patterns can thus reveal the complexity of population spatial structures and the ecological relationship between plants and the environment. Meanwhile, the point pattern analysis method can digitize the position of individual plants, and then a spatial two-dimensional point map can reveal the spatial distribution patterns of various environmental factors at different scales [38,39]. Therefore, the relationships of the spatial distribution patterns of herbaceous plants with environmental factors, plant characteristics and population characteristics can be better clarified through the fine-scale point pattern analysis method [40–42].

Consequently, we analyzed the fine-scale spatial distribution patterns of the clonal plant, *Phragmites australis*, across the environmental gradients in Dunhuang Yangguan Wetland, as a representative of the salt marsh wetlands in northwestern China. The main aims of this study were: (1) to compare the spatial distribution patterns of *Phragmites australis* under environmental heterogeneity, and (2) to analyze whether the spatial distribution pattern is affected by the soil environment and the coevolutionary relationships among the population and clonal characteristics for the fine-scale spatial distribution pattern of *Phragmites australis* in salt marsh wetlands. The results of this study can help reveal the survival strategies of clonal plants and provide a theoretical basis for the protection and management of inland salt marsh wetland ecosystems in extreme environments, such as arid areas.

## 2. Materials and Methods

### 2.1. Overview of the Study Area

The study area is located in the Dunhuang Yangguan National Wetland Reserve, Gansu Province, northwest China (93°53'–94°17' E, 39°39'–40°05' N), with a total area of $8.82 \times 10^4$ hm$^2$ and an altitude of 1150 to 1500 m, as shown in Figure 1. It is a typical salt marsh wetland in the extremely arid desert area of northwest China and is characterized by a warm temperate and arid climate, with a large temperature difference between day and night. The annual average temperature is 9.3 °C. The annual average precipitation is only 39.9 mm, and the annual average evaporation is as high as 2465 mm. The soil in the study area shows a relatively high salinity and is mainly composed of aeolian sand soil, swamp soil, brown desert soil and meadow soil. The vegetation is dominated by temperate desert plants and wetland plants with strong tolerance to drought, cold and salt, including *Phragmites australis*, *Nitraria tangutorum*, *Salicornia europaea*, *Alhagi sparsifolia*, *Lycium ruthenium*, *Glycyrrhiza uralensis*, and *Apocynum venetum*.

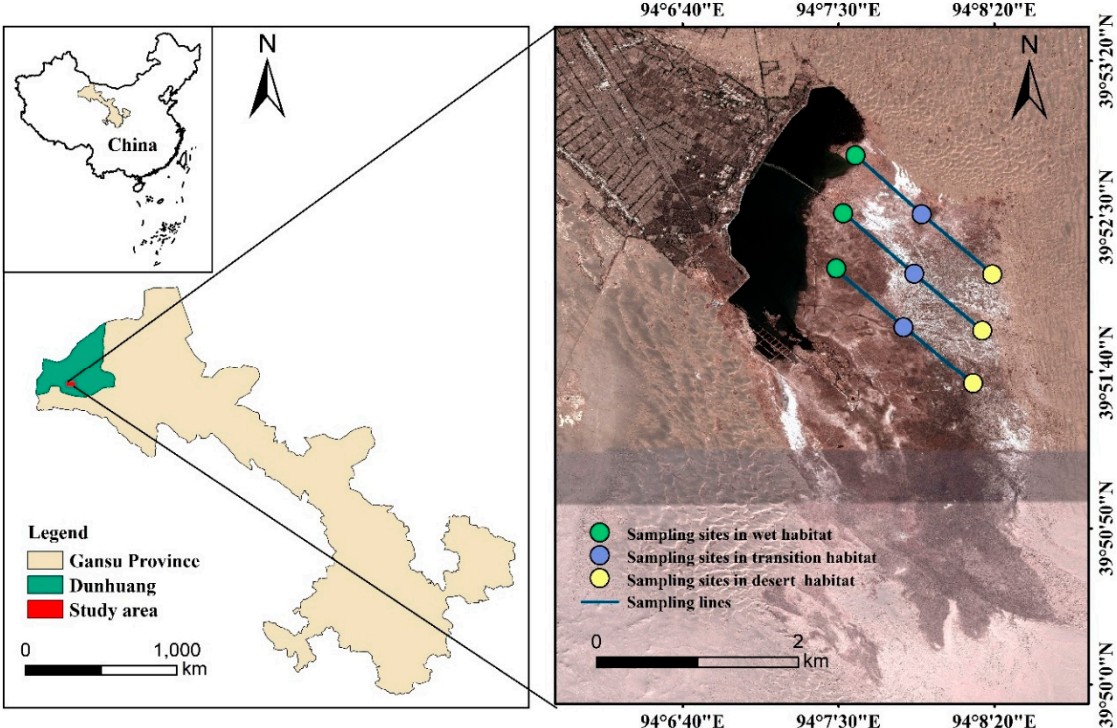

**Figure 1.** Study area and sampling point distribution.

### 2.2. Plant Sampling and Measurement

Three parallel sampling lines, with a spacing of 400 m, were set up along the direction from the wetland to the desert on the basis of a comprehensive inspection of the plants and soil in the study area in July 2018. Then, according to the distance to the reservoir and the *Phragmites australis* cover, three sampling gradients were set up for each sampling line: the wet zone (500 m, cover: 70–95%), the transitional zone (1500 m, cover: 45–70%) and the desert zone (2500 m, cover: 15–45%). Three sampling plots, with a size of 2 m × 2 m were randomly set up for each gradient, and the corresponding altitude and geographical location of each plot were recorded using GPS.

The point pattern characteristics of the *Phragmites australis* population were investigated by the grid method. Each 2 m × 2 m plot was divided into four 1 m × 1 m subplots, each of which consisted of 100 grids of 10 cm × 10 cm [43]. The southwest corner of the plot was taken as the coordinate origin, with the X axis extending from west to east, and the Y axis from south to north. The relative position

and ground projection of each *Phragmites australis* clonal ramet in the plot were recorded. At the same time, the population characteristics of *Phragmites australis*, including the population cover, frequency, density, plant height and aboveground biomass in each 1 m × 1 m subplot, were investigated.

Based on the characteristics of the cloned plant community, three whole *Phragmites australis* plants were randomly selected from each 2 m × 2 m quadrat. Herein, all the aboveground and underground parts and the connected clonal ramets of the selected plants were obtained, according to the clonal module collection method, with full digging, using "tracking and excavation", and were brought back to the laboratory to measure their stem diameter, node number, ramet number, rhizome internode length (length between two sections of the rhizomes underground), spacer length (distance between the clone ramet) and branch angle of the rhizomes (angle between the branches and rhizomes) [44].

The soil sampling points correspond to the sampling points of the plants. In each gradient, the soil was sampled every 10 cm layer to a depth of 100 cm. The surface, middle and deep soil layers were obtained by mixing soils at depths of 0–30 cm, 30–60 cm, and 60–100 cm, respectively. The soil samples were then brought back to the laboratory. The impurities were removed from the soil samples, and the physical and chemical properties of the soil were determined after air drying, grinding and sieving. The soil water content was determined by the drying method. The soil bulk density was measured by ring shear testing (50 cm$^3$). The soil salinity was determined by the water bath evaporation method. The pH value of the soil was determined by a PHS-2F digital pH meter (INESA Scientific InstrumentCo., Ltd., Shanghai, China).

## 2.3. Data processing

### 2.3.1. Point Pattern Analysis

Ripley's K function is commonly used to characterize spatial point patterns, based on the distribution of the distances between pairs of points. The univariate Ripley's K function makes it possible to test the hypothesis of the null model for complete spatial randomness (CSR) and determine whether the spatial distribution pattern of a population is aggregated, random or regular. The simplified function equation is as follows [45]:

$$K(d) = \lambda^{-1}(d > 0),\tag{1}$$

where *d* is an arbitrary vector of distance, and $\lambda$ is calculated by dividing the number of plants by the area of the study region (i.e., $\lambda = n/A$), which is an unbiased estimator of the intensity of the process.

$$K(d) = \frac{A}{n^2} \sum_i^n \sum_j^n \frac{I_d(u_{ij})}{W_{ij}} (i \neq j, d > 0),\tag{2}$$

In practice, equation (2) is often used for estimation. In equation (2), $u_{ij}$ is the distance between point *i* and point *j*. If $u_{ij} > d$, $I_d(u_{ij}) = 0$; and if $u_{ij} \leq d$, $I_d(u_{ij}) = 1$. $W_{ij}$ is the isotropic edge correction function.

$$L(d) = \sqrt{\frac{K(d)}{\pi}} - d,\tag{3}$$

The transformed *L(d)* is usually used to represent the distribution type of point events under the distance scale *d*, and the *L(d)* value indicates the intensity of the spatial distribution pattern of a population. $\sqrt{\frac{K(d)}{\pi}}$ can stabilize the variance and has a linear relationship with *d* under a random distribution. If $L(d) = 0$, it is a random distribution; if $L(d) < 0$, it is a regular distribution; and if $L(d) > 0$, it is an aggregated distribution.

Monte Carlo simulation of the stochastic process underlying the specific null model was used for the construction of confidence limits in Programita 2014 (Wiegand and Moloney). Each simulation generated a *L(d)* value. We calculated approximate $n/(n + 1) \times 100\%$ confidence limits from the highest and lowest values of these functions, taken from *n* simulations of the null model (e.g., *n* = 99 simulations

correspond to a 99% confidence limit). The confidence limit delimits the region of acceptance of the hypothesis of a random spatial distribution pattern. *d* was taken as the abscissa, and the upper and lower envelope traces were taken as the ordinate. The *L*(*d*) value was calculated from the actual distribution data of the *Phragmites australis* population. If *L*(*d*) is within the envelopes, this means that plants are randomly distributed at this scale. If *L*(*d*) is above the upper envelope trace, the distribution of plants is aggregated. If *L*(*d*) is below the lower envelope trace, the distribution of plants is regular [38,46].

### 2.3.2. Data Treatment and Statistical Analysis

The collation and analysis of point pattern data were carried out using Excel 2013 and Programita 2014 software, respectively. On the basis of the Kolmogorov–Smirnov test, the differences in the population characteristics, clonal characteristics and soil environmental factors were analyzed by one-way ANOVA and the least-significant difference (LSD) in SPSS 22.0 (IBM, New York, NY, USA). The significance level was set to $p < 0.05$. The charts were produced by Origin 10.4 and Microsoft Excel 2013.

Redundancy analysis (RDA) is a sorting method and an extension of multi-response regression analysis [47]. In this analysis, data table *Y* contained the value of *L*(*d*), clonal characteristic indicators, and population characteristic indicators, and data table *X* contained the soil environmental factors. After the detrended correspondence analysis (DCA), the lengths of gradient (LGA) for the sorting axis were less than 3, which indicated that the data had a good linear response and were suitable for using RDA. Therefore, the relationships of the intensity of the spatial distribution pattern with the soil environmental factors, population characteristics and clonal characteristics were analyzed by the method of RDA in CANOCO 5.0 (Microcomputer Power, New York, USA).

Based on the linear, independent, and normal distributions, as well as the equal variance of variables, the relationships between the *L*(*d*) value and soil environmental factors of each soil layer were analyzed by the method of simple linear regression. The equation is as follows [48]:

$$y = \alpha + \beta x, \tag{4}$$

In formula (4), $\alpha$ is the intercept, and $\beta$ is the slope. Equation (4) is usually used to characterize the degree to which variable *y* is constrained by variable *x*.

## 3. Results

### 3.1. Analysis of the Soil Environmental Factors in the Study Area

As shown in Table 1, the average soil water content (0–100 cm) in the study area decreased ($p < 0.05$) from the wet zone (20.49 ± 0.66%) to the transitional zone (12.54 ± 0.09%) and to the desert zone (3.33 ± 0.03%). The average soil bulk density and soil salinity also decreased ($p < 0.05$) from the wet zone (bulk density: 1.49 ± 0.01 g/cm$^3$, salt content: 1.89 ± 0.04%) to the transitional zone (bulk density: 1.25 ± 0.01 g/cm, salt content: 1.64 ± 0.04%) and to the desert zone (bulk density: 1.2 ± 0.02 g/cm$^3$, salt content: 1.38 ± 0.09%). The trends of the soil water content, bulk density and salinity at the surface, middle and deep soil layers with the environmental gradient were similar to those of the above corresponding average values. It should be noted that the environmental gradient had no significant influence on the average soil pH and the pH of each soil layer ($p > 0.05$). However, the pH values were greater than 8 under the three gradients, indicating that the soil was weakly alkaline in the study area.

**Table 1.** Soil environmental factors (0–100 cm) across the environmental gradients (mean ±SE).

| | Layer Depth | Wet Zone | Transition Zone | Desert Zone |
|---|---|---|---|---|
| Soil water content (%) | 0–30cm | 17.34 ± 0.68 [a] | 6.49 ± 0.20 [b] | 1.54 ± 0.10 [c] |
| | 30–60cm | 19.8 ± 0.77 [a] | 12.81 ± 0.19 [b] | 4.42 ± 0.23 [c] |
| | 60–100cm | 24.32. ± 0.57 [a] | 19.63 ± 0.25 [b] | 4.04 ± 0.30 [c] |
| | 0–100cm | 20.49 ± 0.66 [a] | 12.54 ± 0.09 [b] | 3.33 ± 0.03 [c] |
| Soil bulk density (g/cm³) | 0–30cm | 2.15 ± 0.14 [b] | 2.74 ± 0.10 [a] | 2.55 ± 0.11 [a] |
| | 30–60cm | 1.36 ± 0.16 [b] | 1.22 ± 0.17 [b] | 1.80 ± 0.10 [a] |
| | 60–100cm | 0.62 ± 0.06 [c] | 0.96 ± 0.02 [b] | 1.31 ± 0.06 [a] |
| | 0–100cm | 1.2 ± 0.02 [c] | 1.25 ± 0.01 [b] | 1.49 ± 0.01 [a] |
| Soil pH | 0–30cm | 8.31 ± 0.07 [a] | 8.19 ± 0.03 [a] | 8.28 ± 0.03 [a] |
| | 30–60cm | 8.24 ± 0.03 [a] | 8.33 ± 0.10 [a] | 8.27 ± 0.03 [a] |
| | 60–100cm | 8.13 ± 0.01 [b] | 8.31 ± 0.09 [a] | 8.28 ± 0.02 [a] |
| | 0–100cm | 8.22 ± 0.03 [a] | 8.28 ± 0.04 [a] | 8.28 ± 0.02 [a] |
| Soil salinity content (%) | 0–30cm | 1.24 ± 0.01 [b] | 1.04 ± 0.01 [a] | 1.26 ± 0.01 [a] |
| | 30–60cm | 1.15 ± 0.03 [c] | 1.36 ± 0.02 [b] | 1.56 ± 0.01 [a] |
| | 60–100cm | 1.21 ± 0.01 [c] | 1.34 ± 0.02 [b] | 1.65 ± 0.01 [a] |
| | 0–100cm | 1.38 ± 0.09 [c] | 1.64 ± 0.04 [b] | 1.89 ± 0.04 [a] |

Different lowercase letters (a, b, c) indicate significant discrepancies between gradients ($p < 0.05$). The same is the case for all subsequent tables.

### 3.2. Analysis of the Clonal Characteristics of Phragmites australis

With the intensification of environmental stress from the wet zone to the desert zone, the stem diameter, node number and ramet number of *Phragmites australis* showed a significant downward trend, while the rhizome internode length, the spacer length and the branch angle showed a significant upward trend (Figure 2). Specifically, the ramet number followed an order of the wet zone (5 ± 0.60) > the transitional zone (3 ± 0.17) > the desert zone (2 ± 0.17); the rhizome internode length followed an order of the desert zone (11.83 ± 0.86 cm) > the transition zone (7.75 ± 0.17 cm) > the wet zone (5.49 ± 0.23 cm); the spacer length followed an order of the desert zone (83.73 ± 6.36 cm) > the transition zone (54.36 ± 1.70 cm) > the wet zone (33.63 ± 1.65 cm); and the branch angle followed an order of the desert zone (76.00 ± 8.43°) > the transitional zone (49.67 ± 2.74°) > the wet zone (41.67 ± 4.41°). From the perspective of statistics, the clonal characteristics, such as the node number, rhizome internode length and spacer length, were significantly different across the environmental gradients ($p < 0.05$).

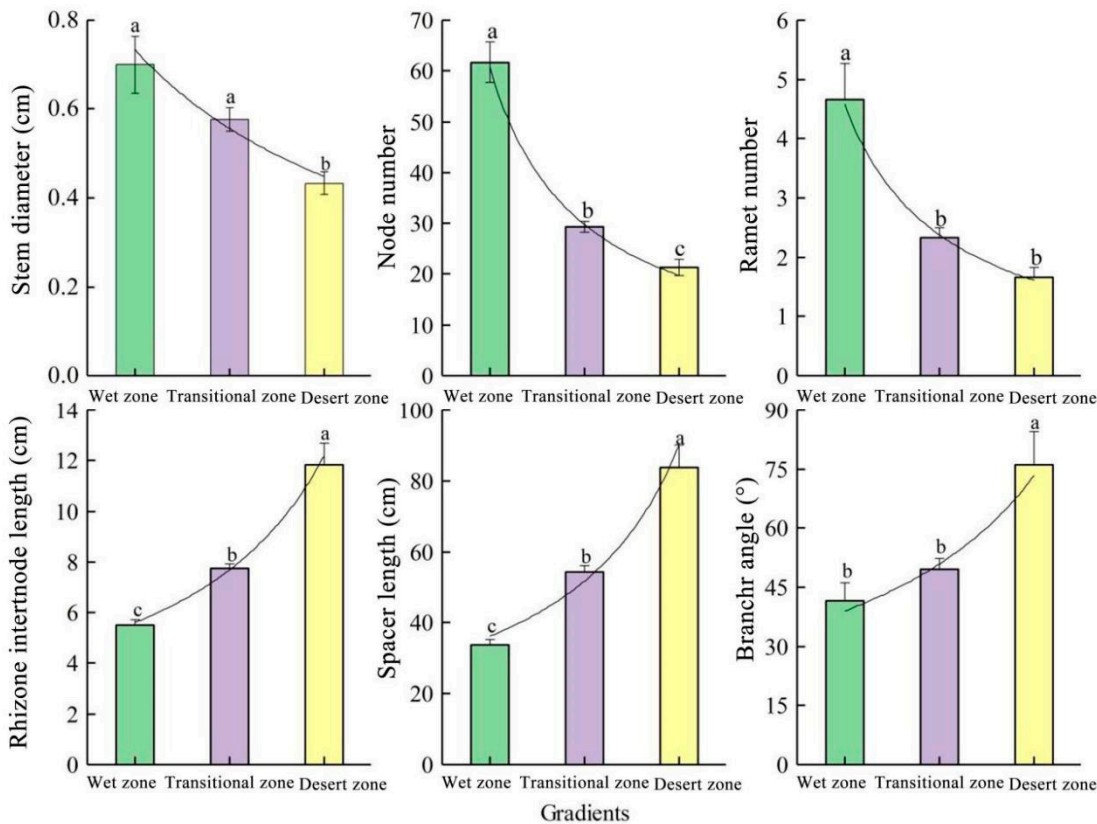

**Figure 2.** Clonal characteristics of *Phragmites australis* across the environmental gradients (mean ± SE). Different lowercase letters indicate significant discrepancies between zones ($p < 0.05$).

### 3.3. Analysis of the Population Characteristics of Phragmites australis

The population characteristics of *Phragmites australis* changed significantly from the wet zone to the desert zone (Table 2). The average cover, frequency and density of the *Phragmites australis* population all followed an order of the wet zone (cover: 86.67 ± 2.20%, frequency: 85.00 ± 1.44%, density: 94.33 ± 3.93 plants/m$^2$) > the transitional zone (cover: 71.67 ± 2.20%, frequency: 68.33 ± 2.20%, density: 51.33 ± 3.03 plants/m$^2$) > the desert zone (cover: 20.00 ± 1.44%, frequency: 20.00 ± 0.01%, density: 28.00 ± 1.04 plants/m$^2$). The average plant height decreased from the wet zone (157.23 ± 11.07 m) to the transitional zone (136.37 ± 3.74 m) and to the desert zone (68.00 ± 5.11 m). In addition, the average biomass decreased significantly from the wet zone (560.01 ± 62.01 g/m$^2$) to the desert zone (98.47 ± 42.51 g/m$^2$).

**Table 2.** Population characteristics of *Phragmites australis* across the environmental gradients (mean ±SE).

| Biological Characteristics | Wet Zone | Transition Zone | Desert Zone |
| --- | --- | --- | --- |
| Cover (%) | 86.67 ± 2.20 [a] | 71.67 ± 2.20 [b] | 20.00 ± 1.44 [c] |
| Frequency (%) | 85.00 ± 1.44 [a] | 68.33 ± 2.20 [b] | 20.00 ± 0.01 [c] |
| Density (plants/m$^2$) | 94.33 ± 3.93 [a] | 51.33 ± 3.03 [b] | 28.00 ± 1.04 [c] |
| Height (cm) | 157.23 ± 11.07 [a] | 136.37 ± 3.74 [a] | 68.00 ± 5.11 [b] |
| Biomass (g/m$^2$) | 560.01 ± 62.01 [a] | 316.18 ± 13.56 [b] | 98.47 ± 42.51 [c] |

Different lowercase letters (a, b, c) indicate significant discrepancies between gradients ($p < 0.05$).

### 3.4. Analysis of the Fine-Scale Spatial Point Pattern of Phragmites australis

The *Phragmites australis* population showed different spatial distribution patterns on different scales in the same gradient (Figure 3). In the wet zone, the population showed a random distribution

on scales of 78–89 cm, an aggregated distribution on scales of 6–77 cm, and a regular distribution on scales of 0–5 cm (d ≥ 90 cm). In the transitional zone, the population pattern appeared as a random distribution on scales of 10–12 cm (d ≥ 66 cm), an aggregated distribution on scales of 13–65 cm, and a regular distribution on scales of 0–9 cm. In the desert zone, the population showed a random distribution on scales of 12–18 cm and 49–90 cm, an aggregated distribution on scales of 19–48 cm, and a regular distribution on scales of 0–11cm (d ≥ 91 cm). From the wet zone to the desert zone, the spatial distribution pattern of the *Phragmites australis* population was generally transformed from an aggregated distribution to a random distribution with the increase of environmental stress.

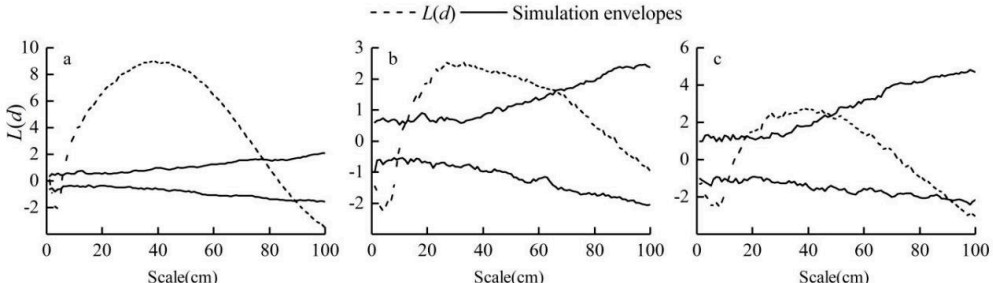

**Figure 3.** Point pattern characteristics of *Phragmites australis* populations across the environmental gradients ((**a**): the wet zone, (**b**): the transition zone, (**c**): the desert zone).

*3.5. Relationships of the Spatial Distribution Pattern of Phragmites australis with the Environmental Factors, Population Characteristics and Clonal Characteristics*

*Phragmites australis'* spatial distribution intensity was significantly affected by population characteristics, clonal characteristics and soil environmental factors (Figure 4). There were significant positive correlations ($p < 0.01$) between the $L(d)$ values and population characteristics, including the cover (R = 0.739), frequency (R = 0.758), density (R = 0.922), height (R = 0.677) and aboveground biomass (R = 0.803). The $L(d)$ values were also found to be positively correlated with the clonal characteristics, including the node number (R = 0.897), ramet number (R = 0.721) and stem diameter (R = 0.572). Besides, the $L(d)$ values were found to be positively correlated with the soil water content (R = 0.885). Notably, there was a significant negative correlation ($p < 0.05$) between the $L(d)$ values and clonal characteristics, including the rhizome internode length (R = −0.755), spacer length (R = −0.779) and branch angle (R = −0.446). The $L(d)$ values were also negatively correlated with the soil environmental factors, including the soil salinity (R = −0.804), bulk density (R = −0.714), and pH (R = −0.446).

In order to further explore the driving force of the population spatial distribution pattern of *Phragmites australis*, the relationships between the *L(d)* and soil environmental factors at different depths were analyzed (Figure 5). The results reveal a significant positive correlation between the *L(d)* and water content in the surface, middle and deep soil layers (R=0.944, 0.882, 0.765, respectively, $p < 0.01$). Besides, the *L(d)* was found to be negatively correlated with the bulk density of the middle and deep soil layers (R = −0.851, −0.818, $p < 0.01$), the pH of the deep soil layer (R = −0.57, $p < 0.01$) and the salinity of the surface and deep soil layers (R = −0.489, −0.846, $p < 0.01$).

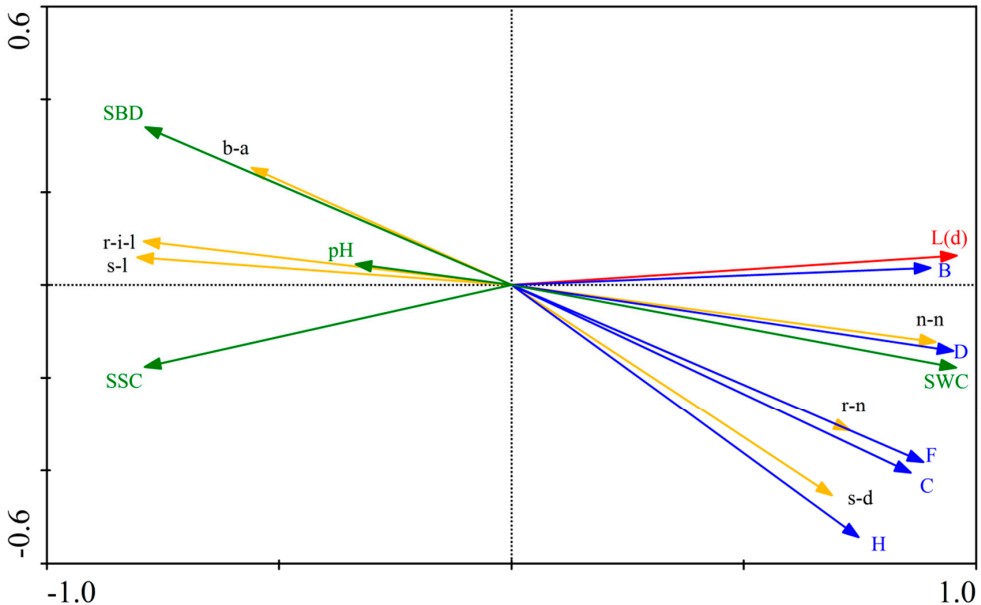

**Figure 4.** Relationships of the *L*(*d*) of the *Phragmites australis* population with the clonal characteristics, population characteristics and soil environmental factors (0-100cm), based on redundancy analysis. SWC: soil water content, SBD: soil bulk density, SSC: soil salinity content, C: cover, F: frequency, D: density, H: plant height, B: biomass, s-d: stem diameter, n-n: node number, r-n: ramet number, r-i-l: rhizome internode length, s-l: spacer length, b-a: branch angle.

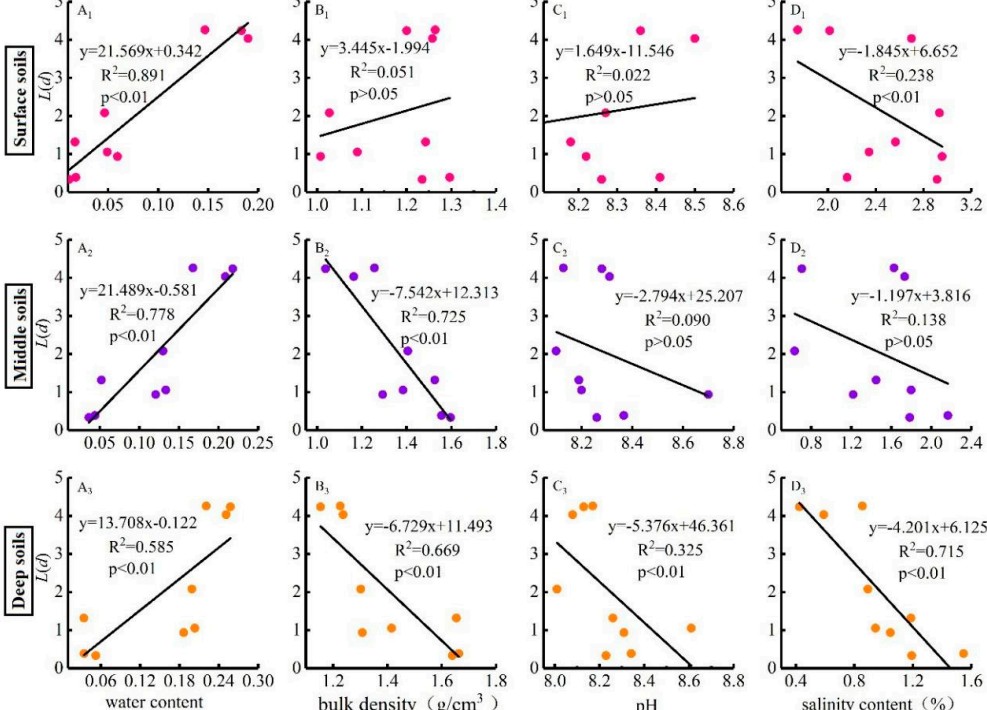

**Figure 5.** Relationships of the *L*(*d*) of the *Phragmites australis* population with the soil environmental factors of each layer (0–30 cm, 30–60 cm, and 60–100 cm), based on simple linear regression. A: Soil water content, B: Soil bulk density, C: Soil pH, D: Soil salinity content, Subscript 1: surface soils (0–30 cm), Subscript 2: middle soils (30–60 cm), Subscript 3: deep soils (60–100 cm).

## 4. Discussion

Plants can change their phenotypic characteristics to increase environmental adaptability under habitat heterogeneity [49]. Such an adaptation mechanism leads to the patchiness of plant population distributions. It has been noted that plant populations show different distribution patterns when investigated at different scales [50,51].

Our study found that the spatial distribution pattern of the *Phragmites australis* population tends to be transformed from an aggregated distribution to a non-aggregated distribution (random and regular distribution) from the wet to the desert zone. Specifically, an aggregated distribution with high $L(d)$ values is the major spatial distribution pattern of the *Phragmites australis* population in the wet zone. In the desert zone, however, a non-aggregated distribution with low $L(d)$ values is the major spatial distribution pattern of the *Phragmites australis* population. *Phragmites australis* achieves a high-intensity spatial distribution pattern in habitats with superior resource and environmental conditions. These conditions are beneficial to their colonization, allowing them to achieve a large ramet number to reach more resources and enhance their overall competitiveness by increasing their population size [45]. In contrast, the intensity of the spatial distribution pattern of *Phragmites australis* is low in habitats with poor resource and environmental conditions. An aggregated distribution helps plants to obtain an advantage in spatial competition, while random and regular distributions are beneficial for reducing the number of inferior individuals and improving the overall fitness of the population [52,53]. Through clonal resource integration and foraging behavior, clonal plants can achieve the optimal life history strategy, based on their spatial distribution patterns. In regions with good resource conditions, *Phragmites australis* distributes more nutrients to the formation of aboveground ramets and thus achieves a high population density, high plant height, high aboveground biomass, large stem diameter, large node number, large ramet number, short rhizome internode length, short spacer length and small branch angle, indicating the "Phalanx" of clonal architecture. In regions with poor resource and environmental conditions, *Phragmites australis* allocates more nutrients to the formation of underground rhizome and thus achieves a small ramet number, small node number, small stem diameter, long rhizome internode length, long spacer length, and large branch angle, indicating the "Guerilla" of clonal architecture [54,55]. With increasing environmental stress from the wet zone to the desert zone, the clonal architecture changes from "Phalanx" to "Guerilla", which leads to changes in the spatial distribution pattern of the *Phragmites australis* population from aggregation to non-aggregation. The research results are consistent with the phenomenon of an increasing non-aggregated distribution of *Kobresia tibetica* and *Suaeda corniculata* in northwest China [56,57].

There were no other species present in the sampling plots. Therefore, the spatial distribution pattern of a plant population is formed by its biological characteristics and the environment in study area. On the one hand, the spatial distribution pattern of a plant population is closely related to the biological characteristics of the plant species itself [58,59], including the diversity characteristics of a plant population and community, life history strategies, regeneration and reproduction patterns, intermediate and intraspecific competition, and inter-clonal trade-off. On the other hand, the spatial distribution pattern of a population is also deeply affected by external environmental factors [60,61], including the soil temperature, humidity, salinity, nutrition, and so on.

From the wet to the desert zone, the population characteristics, including the cover, frequency, density, height and biomass of *Phragmites australis* show significant downward trends and significant positive correlations with the intensity of the spatial distribution $L(d)$. From the wet to the desert zone, the clonal characteristics, including the node number, ramet number, and stem diameter, of *Phragmites australis* show significant downward trends and significant positive correlations with $L(d)$, while the clonal characteristics, including the rhizome internode length, spacer length and branch angle, show upward trends and significant negative correlations with $L(d)$. It is thus inferred that population and clonal characteristics play a decisive role in the formation of the spatial distribution patterns of *Phragmites australis*. With increasing resources, the clonal plants allocate more nutrients to the formation of aboveground parts, and the population density, cover, biomass growth, spatial

distribution intensity and aggregation scale are significantly increased to improve the utilization rate of regional resources and occupation [62]. When subjected to environmental stress, clonal plants allocate more nutrients to the formation of underground parts, which may be far away from the ortet in order to find necessary resources, leading to non-aggregated spatial distribution patterns [4,63]. These results are similar to those obtained for *Leymus chinensis* in Inner Mongolia, China, *Melica przewalskyi* in Gansu province, China, and *Elymus repens* in Southeast France [58,64,65].

Soil water, salinity and pH are the main factors affecting vegetation composition and population distribution in arid regions with little precipitation and high evaporation. This is also confirmed by our study, showing that $L(d)$ is positively correlated with soil water and significantly negatively correlated with soil salinity and pH. Specifically, $L(d)$ is significantly related to the water content at the surface, middle and deep soil layers, the salinity of the surface and deep soil layers and the pH of the deep soil layer. A sufficient water supply promotes the photosynthesis of plant leaves and the absorption of nutrients by plant roots, which is beneficial for plant growth, energy metabolism and nutrient storage. Therefore, the water content in each soil layer has a positive effect on the spatial distribution pattern of the *Phragmites australis* population in the salt marsh wetlands of arid regions. The increased aggregation and high $L(d)$ value, with a high soil water content, increased randomness, and low $L(d)$ value, with a low soil water content, reveal that *Phragmites australis* tends to monopolize water resources in good habitats, and avoid self-harm and seek new water sources in poor habitats. Therefore, soil water is an important driving force for the spatial distribution pattern of plants. Control experiments on plant responses to water stress show that the growth rates of *Eucalyptus microtheca* and *Coriaria nepalensis* decrease, and their spatial distribution patterns change, under drought conditions [66,67]. In addition, plants growing in soils with a high salt content and pH value tend to excessively accumulate ions, like $Na^+$ and $Cl^-$, which break the original osmotic balance of cells, hinder their normal physiological and ecological functions, and restrict plant growth and reproduction [68]. Meanwhile, since the roots of *Phragmites australis* in salt marsh wetlands, accustomed to absorbing soil mineral nutrients, are mainly distributed in deep soil layers, the spatial distribution pattern of *Phragmites australis* is more closely related to the salinity and pH of deep soil layers. Both field and control experiments show that excessive salt can inhibit plant germination, reproduction and growth, and even cause plant death [69]. From the wet zone to the desert zone, the soil water content decreases, and the soil salinity and alkalinity increases. Such a trend of environmental stress drives the allocation of resources from aboveground parts to underground parts and promotes the transformation of clonal architecture from "Phalanx" to "Guerilla", which leads to a decrease in population diversity and stability. Finally, the ecological adaptation strategy of *Phragmites australis* gradually changes from an aggregated distribution to a non-aggregated distribution.

## 5. Conclusions

The spatial distribution pattern of a plant population can reflect the ecological adaptation strategy of the species in a heterogeneous environment. The point pattern method was used here to analyze the fine-scale spatial distribution pattern of the clonal plant, *Phragmites australis*, in the salt marsh wetland of Dunhuang, China. The results show that the spatial distribution pattern of *Phragmites australis* changes from aggregation to non-aggregation (random and regular distribution) as the environmental stress increases from the wet zone to the desert zone. The change of the spatial distribution pattern is probably due to the adaptive foraging behavior and physiological integration of clonal plants. This study contributes to the understanding of the spatial distribution pattern and ecological adaptation mechanism of clonal plants in heterogeneous environments. Additionally, this study also verified that the spatial distribution pattern of a plant population is closely related to soil environmental factors (especially the water content in each soil layer and the salinity and alkalinity of the deep soil layer), clonal characteristics and population characteristics. The results suggest that we should pay more attention to plants' resource allocation strategies and clonal adaptation traits, as well as the effects of

soil environmental factors, when developing measures for ecosystem protection and restoration in sensitive and fragile inland salt marsh wetlands.

**Author Contributions:** L.J. and F.L. collected experimental data, analyzed, and wrote the manuscript. X.L., S.W. and Y.Z. participated equally in conceptualization, data analysis, and manuscript preparation and review. All authors have read and agreed to the published version of the manuscript.

**Funding:** This research was funded by the National Natural Science Foundation of China (No. 41361010 and 41861006), the Scientific Research Program of Higher Education Institutions of Gansu Province (No. 2018C-02) and the Research Ability Promotion Program for Young Teachers of Northwest Normal University (NWNU-LKQN2019-4).

**Acknowledgments:** We especially thank Gansu Dunhuang Yangguan National Nature Reserve Administration for supporting to our research work. The authors also thank the anonymous referees for helpful comments on the manuscript.

**Conflicts of Interest:** The authors declare no conflicts of interest.

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
