# Peer review of "Fine-Scale Distribution Patterns of Phragmites australis Populations Across an Environmental Gradient in the Salt Marsh Wetland of Dunhuang, China"

_sustainability, doi:10.3390/su12041671_

Round 1
Reviewer 1 Report
An interesting study exploring the spatial patterns associated with a clonal plant along a transect zonation. I think the results are interesting, and is a good application of spatial statistics to explore ecological patterns and processes. Subsequently, I feel this could make a good contribution. However, I have a number of comments that I feel must be addressed prior to publication.
The introduction needs to be more focused and informative as to what you are trying to achieve. The first time I read your study I thought you were actually trying to undertake a completely different research project, and it wasn't until you got to the methods and described Ripley's K that I realised this was the analysis. In part, you use vague terms to describe the same phenomena, e.g., population spatial patterns and spatial distributions are two very different types of analytical procedures within spatial ecology, and you use both of these in the first paragraph. Your second paragraph discusses the differences in scale, which while very interesting, is lacking the methodological grounding from the first paragraph for the reader to really understand the importance of what you're stating. I think you need to revisit the introduction and restructure this so that you clearly outline the big picture issues, provide definitions of all terms, and make sure you're clear on the terms you use describing the processes you wish to describe, as this does go beyond semantics. I think there would be a benefit in introducing the ecology (clonal plants, arid wetlands) first, to set the scene and outline why it is important, then go into the methods and hierarchical scales. L125-129. While you provide references that you follow a method of data collection, what is the bias introduced by randomly selecting these three individuals? How many plants were in each grid? These comments need to be addressed, albeit if only a discussion, as they play a big part in your analysis. Section 2.3.2 needs more explanation, as it is very light. L166 - how did you undertake the analysis of Ripley's K in Excel? Adding 'respectively' to the end of the sentence, if you used Excel for collation, and Programita for ripleys. Similarly, what population characteristics were tested? Why did you only use one-way ANOVAs, what was the assumption that none of the variables interacted with each other. Clarify if the ANOVAs are the same as the simple linear regressions you report in the results. Section 3.4 - You've presented the results at cm level. What was the rationale behind calculating different spatial scales in the data collection phase, if you haven't explored the relationships across different extents? L297-304 - What other species were present in your grids, and did you measure them? What would be the differences or relationships if you included these in the analysis.
Reviewer 2 Report
Here authors aimed to study spatial distribution of clonal plants and how are affected by resource heterogeneity and environmental gradients. I found the authors’ approach and data intestesting to better understand the environmental processes that structure and affect the distribution of clonal plants. In general, I think the focus and the objectives of the present manuscript are interesting for plant ecologists, but I think the manuscript as is presented now needs a little more work. Specifically I think that authors need to strength and sharp the information given in different parts of the manuscript with the aim to give a clearer message to readers.
General comments
From my point of view, I think there is a strong incongruence between the issues presented in the introduction and the results presented by authors. In the introduction authors provide a strong emphasis on the importance to study the spatially-explicit distribution of plants but that part of the manuscript’s justification is only considered in one section of results (specifically, 3.4 Analysis of fine-scale spatial point pattern of Phragmites australis). To solve this incongruence, I suggest to authors to refocus the introduction on environmental variability affecting distribution of plants, whereas the fine-scale patterns could be considered by a single paragraph. For me, the 3.1, 3.2 and 3.3 are better explained by large environmental gradients related to environmental variability (wet, transition and desert zone), whereas the 3.4 section is more related to fine-scale patterns dependent on environmental gradients. As such, I consider that authors are not clearly reflecting the key issues presented in the paper, which should be clearly mentioned (i.e. I consider that the keyword “environmental gradients” have to be included in the title).
Minor comments
L37. Modify here and thoroughtot the text “Spatial patters of plant populations”.
L44. Modify here and thoroughtot the text “Spatial patters of vegetation”.
L45-6. I consider this example not intesesting given is not related to forbs or grasses, clonal plants and/or marshes. There are many bibliography on this issue and I do not know why authors provide details about it. Please delete sentence.
L55. Modify “the position of research object in space” by “the position of individual plants”.
L116. It could be really useful to have a look of the raw maps used for this work (i.e. include maps appendix section).
L118. Phargmites australis is a clonal plant so its presence in a given grid or sampling point do not means that belongs to different individuals. On the other side, it could be possible that a given individual is presented in a single 2 m x 2 m plot. If I understood well, I think this issue key for better understanding the results provided by authors. I suggest authors that such issue should be included in the discussion as a potential caveat of the present manuscript. If that issue is not well understood, I suggest that authors provide more information in material and methods.
Figure 2. Please provide the name of each environmental gradient instead of numbers.
Round 2
Reviewer 1 Report
Revisions have clarified this manuscript, and it reads much clearer, particularly in regard to the overall aims and situation within the wider context. A couple of minor text alterations below, but overall I think you've done a good job of incorporating revisions and clarifying points.
L52 - 'a more fragile ecosystem'
L125-127 - awkward sentence, which I think has been incorrectly split which has disrupted flow.
Table 1 - try to keep on the same page
Response to point 8 - this might be worth stating in the MS that there were no other species present. Or include this alongside discussion on
Author Response
Response to Reviewer 1 Comments
Point 1: L52 - 'a more fragile ecosystem'
Response 1: I am sorry for this expression. We have fully checked and modified the statements in the text (Please see Lines 51-52).
Point 2: L125-127 - awkward sentence, which I think has been incorrectly split which has disrupted flow.
Response 2: Thanks! We have accepted your suggestions and adjusted the sentence structure to make the sentence more fluent and complete (Please see Lines 126-128).
Point 3: Table 1 - try to keep on the same page
Response 3: Many thanks for the constructive comments. We have accepted your suggestions and kept Table 1 on one page in the text (Please see Lines 201-203).
Point 4: Response to point 8 - this might be worth stating in the MS that there were no other species present. Or include this alongside discussion on.
Response 4: I am sorry for not clearly stated in point 8. We have fully accepted the suggestion and added this worthy statement in the Discussion (Please see Lines 309-311).
Reviewer 2 Report
Dear Authors,
Thank you for your effort for improving the present manuscript. I do not have further comments.
Best,
Author Response
Thank you very much for your comments and recognition of our paper.